# Potential of Bacteriocins from *Lactobacillus taiwanensis* for Producing Bacterial Ghosts as a Next Generation Vaccine

**DOI:** 10.3390/toxins12070432

**Published:** 2020-07-01

**Authors:** Sam Woong Kim, Yeon Jo Ha, Kyu Ho Bang, Seungki Lee, Joo-Hong Yeo, Hee-Sun Yang, Tae-Won Kim, Kyu Pil Lee, Woo Young Bang

**Affiliations:** 1Gene Analysis Center, Gyeongnam National University of Science & Technology, Jinju 52725, Korea; swkim@gntech.ac.kr; 2Department of Pharmaceutical Engineering, Gyeongnam National University of Science and Technology, Jinju 52725, Korea; chakfhd@daum.net (Y.J.H.); khbang0095@gntech.ac.kr (K.H.B.); 3National Institute of Biological Resources (NIBR), Environmental Research Complex, Incheon 22689, Korea; metany@korea.kr (S.L.); y1208@korea.kr (J.-H.Y.); moeicdy@korea.kr (H.-S.Y.); 4Department of pharmacology, College of Veterinary Medicine, Chungnam National University, Daejeon 34134, Korea; taewonkim@cnu.ac.kr; 5Department of physiology, College of Veterinary Medicine, Chungnam National University, Daejeon 34134, Korea

**Keywords:** AMPs, antimicrobial activity, bacterial ghost, bacteriocin, *Lactobacillus taiwanensis*

## Abstract

Bacteriocins are functionally diverse toxins produced by most microbes and are potent antimicrobial peptides (AMPs) for bacterial ghosts as next generation vaccines. Here, we first report that the AMPs secreted from *Lactobacillus taiwanensis* effectively form ghosts of pathogenic bacteria and are identified as diverse bacteriocins, including novel ones. In detail, a cell-free supernatant from *L. taiwanensis* exhibited antimicrobial activities against pathogenic bacteria and was observed to effectively cause cellular lysis through pore formation in the bacterial membrane using scanning electron microscopy (SEM). The treatment of the cell-free supernatant with proteinase K or EDTA proved that the antimicrobial activity is mediated by AMPs, and the purification of AMPs using Sep-Pak columns indicated that the cell-free supernatant includes various amphipathic peptides responsible for the antimicrobial activity. Furthermore, the whole-genome sequencing of *L. taiwanensis* revealed that the strain has diverse bacteriocins, confirmed experimentally to function as AMPs, and among them are three novel bacteriocins, designated as Tan 1, Tan 2, and Tan 3. We also confirmed, using SEM, that Tan 2 effectively produces bacterial ghosts. Therefore, our data suggest that the bacteriocins from *L. taiwanensis* are potentially useful as a critical component for the preparation of bacterial ghosts.

## 1. Introduction

*Salmonella enterica* serovar Gallinarum (*Salmonella* Gallinarum) and the avian pathogenic *Escherichia coli* (APEC) are major public health concerns in domestic poultry species causing acute illnesses, such as typhoid fever and colibacillosis, respectively [1,2]. For the prevention of these infections, vaccines traditionally developed using live-attenuated and killed microorganisms have been widely used [3]. Particularly, killed microorganisms remain the most widely used vaccines to control these infectious diseases. This is because they carry no risk of reversion like the live attenuated ones; however, they are less effective in providing immune protection than the live attenuated ones [4,5]. For this reason, recent studies have highlighted the next generation vaccines based on bacterial ghosts [3].

The bacterial ghosts are empty cell envelopes derived from bacteria conserving antigenic characters on their surface and lacking the internal components [3]. Accordingly, they represent a potential platform that not only acts as a safe vaccine but also triggers effective humoral and cellular immune responses [3]. Bacterial ghosts generally have been produced by controlling the expression of the cloned lysis gene *E* of bacteriophage phiX174, which encodes a 91-aa polypeptide with the ability to oligomerize into a transmembrane of the host cell for lysis tunnel formation [3,6]. However, the lysis is occasionally incomplete because of the resistance of the host cell against the gene *E* and thus there is a constant occurrence of surviving bacteria [7,8], leading to the failure of safe vaccine production. Therefore, for complete bacterial lysis, additional antimicrobial agents need to be used, along with the expression system of the cloned lysis gene *E*.

Antimicrobial peptides (AMPs) are antimicrobial agents suitable for the bacterial ghosts because they are known to form pores in bacterial membranes, as does the lysis protein E [9]. AMPs are small peptides ranging from 10 to 40 amino acids in size and have common features, such as cationicity and amphipathicity [9,10]. Although the exact mechanism of their action is not fully understood, AMPs are known to cause microbial membrane damage via pore formation [9]. Accordingly, AMPs have shown an interesting potential for producing bacterial ghosts; for example, the temporin L isolated from frog skin and the model amphipathic peptide (MAP) effectively formed the bacterial ghosts [11,12]. However, their practical application is limited because the chemical synthesis of large amounts of AMPs is not feasible in low unit cost and heterologous expression systems were successful in producing only few AMPs [13,14].

Bacteriocins, functionally diverse toxins secreted by most microbes, are the most promising AMP candidates [15,16], and their production via bacterial culture can accelerate their practical application to the next generation vaccines [17,18]. The ones from lactic acid bacteria are particularly known to possess huge potential as AMPs [19]. In this study, we first report that the cell-free supernatant from *Lactobacillus taiwanensis*, a lactic acid bacterium, effectively forms the ghost of pathogenic bacteria, mediated by AMPs, and we discovered three novel bacteriocins exhibiting significant antimicrobial activity through the genomic analysis of *L. taiwanensis*. Altogether, our study will provide important information that will guide new strategies to effectively produce bacterial ghosts for their practical application.

## 2. Results

### 2.1. Antibacterial Activity of Cell-Free Supernatant from L. taiwanensis

The antimicrobial activity of the cell-free supernatant from *L. taiwanensis* was examined against two Gram-negative bacteria (*Salmonella* Gallinarum and enteropathogenic *Escherichia coli* (EPEC) isolated from fowls), two Gram-positive bacteria (*Bacillus cereus* and *Streptococcus iniae*) and a yeast, *Saccharomyces cerevisiae* (Figure 1). The minimum inhibitory concentrations (MIC50) were determined as 77.1 and 80.1 μg peptides/mL against the two Gram-negative bacteria, *S.* Gallinarum and EPEC, respectively (Figure 1a,c), and as 417.4 and 2214.8 μg peptides/mL against the two Gram-positive bacteria, *B*. *cereus* and *S*. *iniae*, respectively (Figure 1b,c). However, against the yeast, *S*. *cerevisiae*, its MIC50 was determined as 5936.9 μg peptides/mL (Figure 1b,c), exhibiting an extremely low antimicrobial activity against the yeast. These results indicate that the cell-free supernatant from *L. taiwanensis* has significant antimicrobial activities against the pathogenic bacteria. In addition, it exhibits a higher antimicrobial activity against Gram-negative than Gram-positive bacteria.

### 2.2. The Efficient Production of Bacterial Ghosts by Cell-Free Supernatant from L. taiwanensis

To identify how the cell-free supernatant from *L. taiwanensis* exerts its antimicrobial role against the pathogenic bacteria, scanning electron microscopy (SEM) was carried out with *S.* Gallinarum, a pathogenic bacterium. As shown in Figure 2, the scanning electron micrographs revealed that the cell-free supernatant from *L. taiwanensis* effectively caused cellular lysis by damaging the *Salmonella* membrane via pore formation, as is the case for typical AMPs [9], whereas the negative control did not. This suggests that the cell-free supernatant may include AMPs forming efficient bacterial ghosts. Further challenges to identify the AMPs were performed, as reported in the following section.

### 2.3. Characterization of Antimicrobial Substances from the L. taiwanensis Cell-Free Supernatant 

To examine if the cell-free supernatant from *L. taiwanensis* contains proteins or peptides responsible for the antimicrobial activities, proteinase K or EDTA was used, as shown in Figure 3 and Figure 4. The cell-free supernatant treatment with proteinase K led to the degradation of its proteins and peptides (Figure 3a), resulting in a considerable decrease in its antimicrobial activity against *S.* Gallinarum, supported by the deactivation of the proteinase K treatment (Figure 3b). Moreover, EDTA, a protease inhibitor, was shown to effectively increase the antimicrobial activity against *S.* Gallinarum (Figure 4). Thus, these results prove that the antimicrobial activities of cell-free supernatant from *L. taiwanensis* are mediated mainly by its proteins or peptides, functioning as AMPs.

When the antimicrobial activity of the supernatants, obtained after the centrifugation of 30%, 50%, or 70% methanol extracts from the cell-free supernatant, was examined against *S.* Gallinarum, one of them, particularly from the 50% methanol extract, exhibited the most superior antimicrobial activity (Figure 5a). Furthermore, they were partitioned using chloroform, then only the water layer was subjected to purification using a Sep-Pak column, and finally the fractions were eluted using 10% to 80% acetonitrile (ACN) and were assessed for antimicrobial activity against *S.* Gallinarum. Although the fractions eluted using 10% to 80% ACN exhibited a lower antimicrobial activity than the flow-through and wash fractions, expected to include hydrophilic peptides, they showed various antimicrobial activities according to the ACN concentrations (Figure 5b), implying that peptides having both hydrophilic and hydrophobic tendencies may function importantly in the antimicrobial activity of the cell-free supernatant from *L. taiwanensis*. Altogether, these data confirm that the cell-free supernatant from *L. taiwanensis* includes various amphipathic peptides responsible for the antimicrobial activity.

### 2.4. Discovery of Novel Bacteriocins by the Genomic Analysis of L. taiwanensis 

To identify the AMPs from *L. taiwanesis*, the whole-genome sequencing of *L. taiwanesis* was performed using the PacBio RS II (Pacific Biosciences, Menlo Park, CA) sequencing platform. After de novo assembly, the genome of *L. taiwanesis* was identified as consisting of a single circular DNA chromosome of 1,782,747 base pairs, a GC content of 37.36%, and no plasmids, and containing 1841 predicted open reading frames (ORFs), 55 tRNAs, and 15 rRNAs (Appendix A). Among the ORFs, seven were identified as encoding proteins homologous with the known AMPs using an NCBI homology BLAST (Figure 6, Appendix A) and as indeed being expressed in the strain *L. taiwanesis* through transcriptomic sequencing (Appendix A). In detail, the four ORFs, orf0100, orf01402, orf01553, and orf01554, were found to be highly homologous with bacteriophage-derived *holin* genes, encoding AMPs such as the phage holin family protein (PHF) and holin 1 to 3, which showed very high identities (>60%) with those from the *Pediococcus* genus (Appendix A). Noticeably, the three ORFs, orf00298, orf00302, and orf01552, were discovered to encode novel bacteriocins because they consist of amino acid sequences having high positives (>50%) with those of bacteriocins such as lacticin, enterocin B, or gramicidin C, despite the low identities (<50%) with those of the already known bacteriocins (Appendix A). Thus, we here designated the novel AMPs as Taiwanencin (Tan) 1 to 3 (Figure 6 and Appendix A).

To confirm whether the three Tans function as AMPs, their synthetic peptides were assessed for antimicrobial activity against a Gram-negative bacterium (EPEC) and a Gram-positive bacterium (*B. cereus*) (Figure 7a,b). Tan 1 exhibited significant antimicrobial activity against only *B. cereus*, a Gram-positive bacterium, whereas Tan 2 and 3 presented significant antimicrobial activity against both Gram-negative and Gram-positive bacteria (Figure 7a,b). Moreover, the scanning electron micrographs revealed that the synthetic Tan 2 effectively caused cellular lysis by damaging the EPEC membrane via pore formation, as did the cell-free supernatant from *L. taiwanensis* (Figure 7c). Collectively, these results suggest that *L. taiwanensis* produces and secretes diverse AMPs, such as Tan 1, 2, and 3, thus forming efficient bacterial ghosts.

## 3. Discussion

*L. taiwanensis* was first discovered in silage as a homofermentative lactic acid bacterium [20], but until now there existed no studies on the antimicrobial substances produced by it, even though it belongs to the genus *Lactobacillus*, which produces various antimicrobial substances, such as bacteriocins [19]. In this study, we first investigated the antimicrobial substances from *L. taiwanensis* and found that the species produces various AMPs, identified to have the potential for producing bacterial ghosts as a next generation vaccine. The cell-free supernatant from *L. taiwanensis* showed antimicrobial activities against various pathogenic bacteria, such as the livestock pathogens *S.* Gallinarum and EPEC, a human foodborne pathogen, *B. cereus*, and a fish pathogen, *S. iniae* (Figure 1), which indicates that it has antimicrobial activities against a broad range of pathogenic bacteria. In particular, it showed higher antimicrobial activity against Gram-negative bacteria than Gram-positive ones (Figure 1), which may presumably be due to a difference in the surface properties between Gram-negative and Gram-positive bacteria [21,22]. For example, Gram-negative bacteria possess a more negatively charged surface than Gram-positive ones, and therefore, they may be more susceptible to positively charged antimicrobial substances, such as silver nanoparticles and typical AMPs [21,22,23]. Thus, further investigations to identify the antimicrobial substances, such as AMPs, from *L. taiwanensis* were performed against Gram-negative pathogenic bacteria.

AMPs are small peptides ranging from 10 to 40 amino acids in size and are known to cause microbial membrane damage via either pore formation through a barrel stave or a toroidal pore mechanism or through a non-pore carpet-like mechanism [9,10]. Our scanning electron micrographs of *S*. Gallinarum showed clearly that the cell-free supernatant from *L. taiwanensis* forms a pore on the *Salmonella* surface (Figure 2), as do typical AMPs [9]. Proteinase K treatment of the cell-free supernatant led to a considerable decrease in its antimicrobial activity against *S.* Gallinarum, with the degradation of its proteins and peptides (Figure 3), and EDTA, a proteinase inhibitor, increased the antimicrobial activity (Figure 4). Thus, these results confirm that the antimicrobial activities of the cell-free supernatant from *L. taiwanensis* are mediated mainly by its proteins or peptides, functioning as AMPs, forming efficient bacterial ghosts.

In addition, typical AMPs have common features, such as cationicity and amphipathicity [9,10], and are known to exhibit optimum antimicrobial activity with 50% hydrophobicity [24]. For example, the precipitate, obtained after the centrifugation of methanol extractions with a very high hydrophobicity, was reported to exhibit very little antimicrobial activity [24]. In this study, the supernatants obtained after the centrifugation of methanol extracts from the *L. taiwanensis* cell-free supernatant showed significant antimicrobial activities against *S.* Gallinarum, and in particular, the supernatant from the 50% methanol extraction exhibited higher antimicrobial activity than the other supernatant (Figure 5a), which supports the fact that the cell-free supernatant from *L. taiwanensis* includes AMPs, with features like cationicity and amphipathicity. Furthermore, when the supernatant from the 50% methanol extraction was subjected to purification using Sep-Pak columns, the fractions eluted using 10% to 80% ACN exhibited various antimicrobial activities against *S.* Gallinarum (Figure 5b), suggesting that the cell-free supernatant from *L. taiwanensis* may include various amphipathic peptides, with both hydrophilic and hydrophobic tendencies, responsible for the antimicrobial activity.

Finally, to identify the AMPs from *L. taiwanesis*, we first performed the whole-genome sequencing of *L. taiwanesis*, which revealed that the seven ORFs encode AMPs such as PHF, holin 1 to 3, and Tan 1 to 3 (Figure 6, Appendix A). The antimicrobial activity of the bacteriophage-derived holin homologs was examined via the cell-free supernatants from *E. coli.* The Top10 strain, harboring each *holin* gene, including its own intact promoter, was cloned into the pGEM^®^-T Easy Vector (Promega, Madison, WI, USA) using primers described in Appendix A, due to the difficulty in peptide synthesis caused by their large size (>80-mer). As illustrated in Appendix A, these homologs exhibited antimicrobial activities against both Gram-negative and Gram-positive bacteria, whereas no antimicrobial activity was detected in the negative control, that is, treatment with the cell-free supernatant from the *E. coli.* Top10 strain harbored only the plasmid vector without the *holin* genes. Presumably, the *E. coli.* Top10 strain harboring each *holin* gene may produce and release holin peptides, functioning as the AMP, into the extracellular media via its own intact promoter and bacterial membrane hole formation. Furthermore, the antimicrobial activities of holin homologs from *L. taiwanesis* are supported by the previous reports, suggesting that the holins from bacteriophages function as AMPs, leading to the lysis of the bacterial host’s cell wall at the end of the lytic cycle by forming pores in the membrane [25,26]. Interestingly, Tan 1 to 3 were first discovered as novel AMPs in this study (Figure 6 and Appendix A) and showed antimicrobial activities against Gram-negative or Gram-positive bacteria (Figure 7a,b). Moreover, SEM revealed that synthetic Tan 2 also effectively caused cellular lysis through damage to the *E. coli* membrane via pore formation (Figure 7c), suggesting that they function as AMPs. 

Meanwhile, AMPs such as bacterocins have been found in a variety of microorganisms in clustered arrangements in specific regions of the chromosome [27,28]. The *L. taiwanensis* genome revealed that *Tan 1*/*Tan 2* and *Tan 3*/*holin 2*/*holin 3* form clusters grouped in a specific area, as observed in previous studies [27,28], but *PHF* and *holin 1* are independently located in specific regions. In particular, *Tan 3*, *holin 2*, and *holin 3* are arranged continuously with similar orientations, implying that they may have an influence on mutual expression. These results confirm that *L. taiwanensis* produces and secretes diverse AMPs, including novel bacteriocins, such as Tan 1 to 3. Altogether, our data indicate that the AMPs from *L. taiwanensis*, especially the three novel bacteriocins, are potentially useful as a critical component to effectively produce bacterial ghosts for the development of next generation vaccines.

## 4. Materials and Methods

### 4.1. Materials

*L. taiwanensis* was supplied from the Korea Collection of Microbial Cultures (National Institute of Biological Resources, the Ministry of Environment, Incheon, Korea) in Korea. *S.* Gallinarum, enteropathogenic *E. coli* JOL418, and *S. iniae*, as susceptible bacteria against AMPs, were obtained from Dr. Jin Hur (Chonbuk National University, Iksan, Korea) and Dr. Tae Sung Jung (Gyeongsang National University, Jinju, Korea). *B. cereus* (ATCC 1178) and *S. cerevisiae* (KCTC 27134) were purchased from KCTC (Korean Collection for Type of Cultures, Daejeon, Korea). *S.* Gallinarum, EPEC, *S. iniae*, and *B. cereus* were incubated at 37 °C, while *S. cerevisiae* was cultured at 30 °C. The three synthetic peptides, Tan 1 to 3, were purchased from Cosmogenetech Inc. (Seoul, Korea).

### 4.2. Analysis of the Minimal Inhibitory Concentration (MIC50)

*L. taiwanensis* was inoculated into an MRS liquid medium and cultured at 37 °C for 24 h. The culture was centrifuged at 2000× *g* for 20 min and the supernatant was collected and then sterilized by filtration through a 0.22 μm filter. The sterilized solution was either used directly for the analysis of antimicrobial activity or fractionated and stored at −80 °C until use.

The antimicrobial activity on a microtiter plate was done by some modification according to the dilution assay of Wiegand et al. [29]. Briefly, to examine the antimicrobial activity on a microtiter plate [13], the serially diluted cell-free supernatant was mixed with the diluted susceptibility test strain (10^6^ CFU/mL) and fresh medium in each well. Then, the microtiter plate was incubated at 37 °C for 24 h, and eventually the MIC50 (the minimal inhibitory concentration required to inhibit the growth of 50% of microorganisms) was determined by measuring the absorbance at 600 nm. Total peptide contents in the cell-free supernatants used in this study were measured by using a Bio-Rad Protein assay kit (Bio-Rad, Hercules, CA, USA), according to the manufacturer’s protocol, and were used to express the MIC50 as a peptide equivalent (μg) per volume (mL) of the sample.

### 4.3. Scanning Electron Microscopy (SEM)

*S.* Gallinarum was incubated with the cell-free supernatant from *L. taiwanensis* culture for 0, 6, and 24 h. The treated *S.* Gallinarum cells were fixed with one volume of 2.5% glutaraldehyde (Sigma-Aldrich, St. Louis, MO, USA) for 24 h at 4 °C. Then, the samples were rinsed with sterile PBS buffer thrice and sequentially dehydrated with graded ethanol (30%, 50%, 70%, 80%, 90%, and 100% (*v/v*); 15 min incubation for each concentration). Finally, the samples were dried at room temperature and sputter-coated with gold for SEM.

### 4.4. Effects of Proteinase K and EDTA Treatments

Proteinase K (100 µg/mL) was added to the cell-free supernatant from *L. taiwanensis*, and was allowed to react for 30 min at 37 °C. EDTA (5–30 mM) was added to the cell-free supernatant from *L. taiwanensis* in a microtiter plate. Thereafter, the proteinase K- or EDTA-treated cell-free supernatant was mixed at MIC50 (77.1 μg peptides/mL) with 10^8^ CFU/mL of *S.* Gallinarum, and was further incubated for 24 h at 37 °C; eventually, its absorbance was measured at 600 nm to examine the antimicrobial activity. Furthermore, the proteinase K-treated cell-free supernatant was concentrated with 20% trichloroacetic acid for SDS-PAGE analysis. Proteins (60 μg) from the concentrated samples were separated via 15% SDS-PAGE, and the protein bands were visualized with Coomassie staining [30].

### 4.5. Methanol Fractionation and Partition by An Organic Solvent

The cell-free supernatant from *L. taiwanensis* was adjusted into 30%, 50%, or 70% methanol concentration, incubated at 4 °C for 1 h, and then centrifuged at 2000× *g* for 20 min. The supernatant was collected, completely dried by concentration under reduced pressure, and suspended in distilled water to generate a 10-fold concentrate of the initial supernatant. Thereafter, the total peptide contents and antimicrobial activity against *S.* Gallinarum were assessed in the concentrated samples to determine the MIC50, as mentioned.

The supernatant from 50% methanol fractionation was mixed with the same volumes of chloroform, and then incubated at 4 °C for 1 h. The reacted solutions were centrifuged at 2000× *g* for 20 min at 4 °C, and recovered into water and organic solvent phases. The recovered water layer was further subjected to purification by a Sep-Pak R18 column, as follows.

### 4.6. Purification by Sep-Pak R18 Column

The water layer in the chloroform partition was mixed with ice-cooled 0.1% trifluoroacetic acid (TFA) in the same amount. The mixed solution was centrifuged at 12,000× *g* for 10 min at 4 °C to remove the precipitate. The collected supernatant was loaded into a Sep-Pak plus C18 column (Waters Corp., Milford, MA, USA). After the free fall of the solution (flow-through), washing was performed twice with 10 bed volumes of 0.05% TFA. The bound peptides were eluted with 20 mL of the solution prepared with 10%, 20%, 30%, 50%, and 80% acetonitrile (ACN) in a 0.05% TFA solution. The eluted solutions were concentrated and adjusted to a final concentration of 10 times the initially used cultural broth, and eventually the total peptide contents and the antimicrobial activity against *S.* Gallinarum were measured to determine the MIC50, as mentioned.

### 4.7. Analysis of Genome

Genomic DNA was extracted from *L. taiwanensis* using the MagAttract HMW DNA Kit (QIAGEN, Germantown, MD, Germany) according to the manufacturer’s instructions. A total of 5 μg of the DNA sample extracted were used as input into the library preparation. The SMRTbell library was constructed with SMRTbell™ Template Prep Kit 1.0 (PN 100-259-100) following the manufacturer’s instructions (Pacific Biosciences, Menlo Park, CA, USA). The small fragments lower than 20 kb of the SMRTbell template was removed using the Blue Pippin Size selection system for a large-insert library. The constructed library was validated by an Agilent 2100 Bioanalyzer. After a sequencing primer was annealed to the SMRTbell template, DNA polymerase was bound to the complex using a DNA/Polymerase Binding kit P6. This polymerase–SMRTbell–adaptor complex was then loaded into SMRT cells. The SMRTbell library was sequenced using one SMRT cell (Pacific Biosciences, Menlo Park, CA, USA) with C4 chemistry (DNA sequencing Reagent 4.0) and 240 min movies were captured for the SMRT cell using the PacBio RS II (Pacific Biosciences, Menlo Park, CA, USA) sequencing platform [31], which produced 82,149 reads and 723,287,462 base pairs after subread filtering.

De novo assembly was conducted using the hierarchical genome assembly process (HGAP, Version 2.3) workflow [32], including consensus polishing with Quiver, an error correction tool (http://bit.ly/pbquiver). As the estimated genome size was 1,838,554 bp and average coverage was 317×, we performed error correction based on the longest of about 30× (150,018,235 bp) seed bases, with the rest being shorter reads, and then assembled this with error-corrected reads. As a result of the HGAP process, we got the results of a 1,838,554 bp N50 contig and a 1,838,554 bp total contig length by the polishing process. Finally, since the bacterial genome was typically circular, we checked the form for the contig using MUMmer 3.5 [33] and identified that the genome of *L. taiwanesis* comprises a single circular DNA chromosome of 1,782,747 bp by trimming one of the self-similar ends for manual genome closure.

Putative gene coding sequences (CDSs) from the assembled contigs were identified using Glimmer v3.02 [34] and open reading frames (ORFs) were obtained. These ORFs were searched using Blastall alignment [http://www.ncbi.nlm.nih.gov/books/NBK1762] against the NCBI non-redundant protein database (nr) for all species. GO annotation was assigned to each of the ORFs by Blast2GO software [35], analyzing the best hits of the BLAST results. Additionally, ribosomal RNAs and transfer RNAs were predicted using RNAmmer 1.2 and tRNAscan-SE 1.4 [36,37].

### 4.8. Statistical Analysis

When significant differences were detected, the mean values were separated by the probability difference option. The results are presented as least square means with standard deviations. Duncan’s multiple range tests (MRT) were employed to verify significant differences (*p* < 0.05) between sample types. All the analyses were performed using the SAS statistical software package (version 9.1, SAS Inst., Inc., Cary, NC, USA), and differences were considered significant at *p* < 0.05.

## Figures and Tables

**Figure 1 toxins-12-00432-f001:**
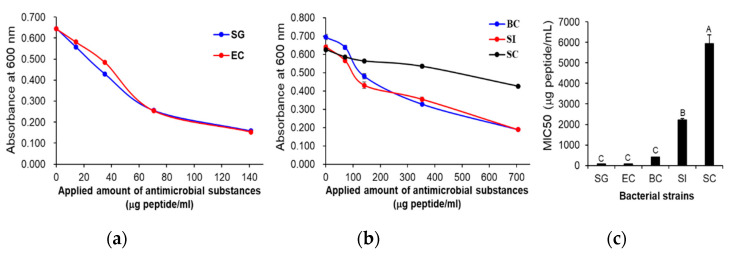
Antibacterial activity of the cell-free supernatant from *L. taiwanensis*. Antimicrobial activities of the cell-free supernatant from *L. taiwanensis* were examined against two Gram-negative bacteria (**a**) and two Gram-positive bacteria and a yeast (**b**), whose MIC50s were determined (**c**). *x*- and *y*-axes indicate the concentration of peptides in the used cell-free supernatant and the absorbances at 600 nm, respectively (**a**,**b**). The *y*-axis and different letters (A, B, and C) in (**c**) represent the MIC50 and significant differences (*p* < 0.05), respectively. SG; *Salmonella* Gallinarum, EC; enteropathogenic *E. coli*, BC; *Bacillus cereus*, SI; *Streptococcus iniae*, SC; *Saccharomyces cerevisiae*.

**Figure 2 toxins-12-00432-f002:**
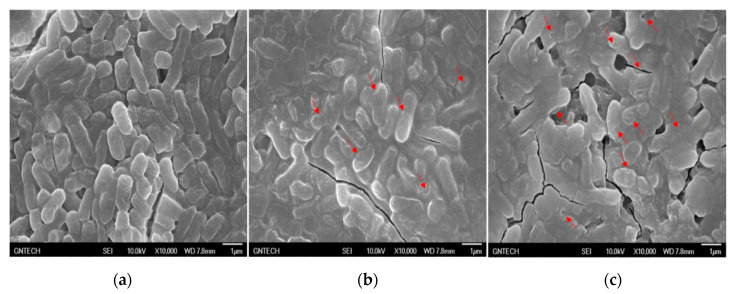
Scanning electron micrographs of *S.* Gallinarum treated with the cell-free supernatants from *L. taiwanensis*. *S.* Gallinarum was treated without (**a**) or with the cell-free supernatant (77.1 μg peptides/mL, MIC50 against *S.* Gallinarum) from *L. taiwanensis* for 6 h (**b**) and 24 h (**c**). The red arrows indicate the pores forming in the *Salmonella* membrane.

**Figure 3 toxins-12-00432-f003:**
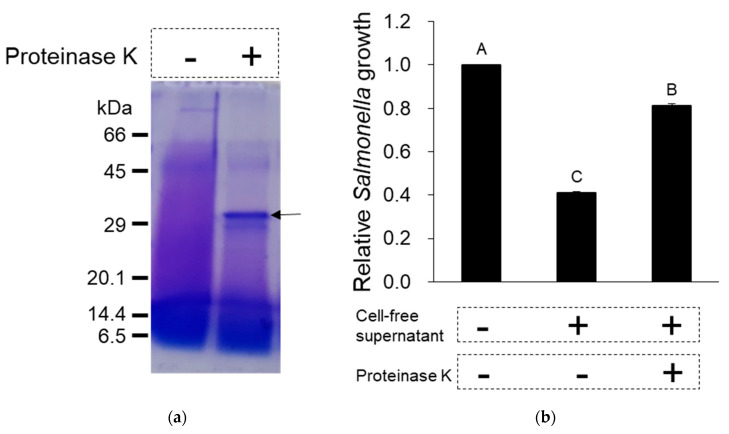
Effect of protease on the antimicrobial activity of the cell-free supernatant from *L. taiwanensis*. Proteinase K (100 µg/mL) was added to the cell-free supernatant (77.1 μg peptides/mL, MIC50 against *S.* Gallinarum) from *L. taiwanensis*. The 15% SDS-PAGE was performed to identify the degradation of proteins and peptides (**a**), and the proteinase K-treated sample was used to examine the antimicrobial activity against *S.* Gallinarum (**b**). The arrow in (**a**) represents degraded proteins or peptides. The *y*-axis and different letters (A, B and C) in the graphs represent the relative growth of *S.* Gallinarum and significant differences (*p* < 0.05), respectively.

**Figure 4 toxins-12-00432-f004:**
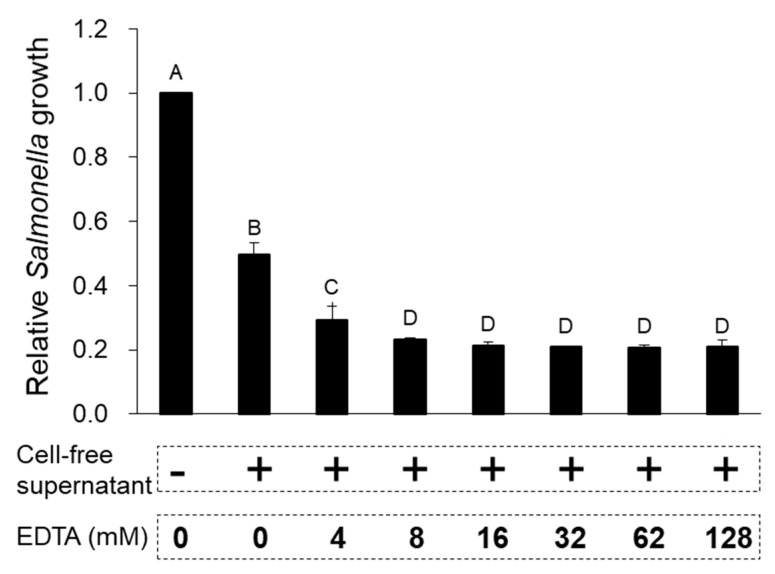
Effect of protease inhibitor on the antimicrobial activity of the cell-free supernatant from *L. taiwanensis*. EDTA, a protease inhibitor, was added to the cell-free supernatant (77.1 μg peptides/mL, MIC50 against *S.* Gallinarum) from *L. taiwanensis* and this sample was used to examine the antimicrobial activity against *S.* Gallinarum. The *y*-axis and different letters (A, B, C and D) in the graphs represent the relative growth of *S.* Gallinarum and significant differences (*p* < 0.05), respectively.

**Figure 5 toxins-12-00432-f005:**
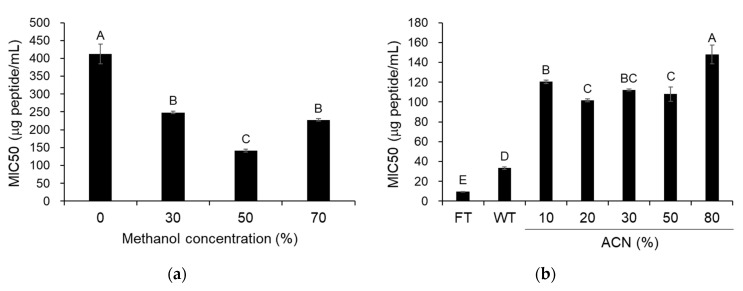
Antibacterial activity of various fractions from *L. taiwanensis* cell-free supernatant. (**a**) The supernatants, obtained after the centrifugation of 30%, 50%, or 70% methanol extracts (M30 to M70) from the cell-free supernatant, were used to examine the antimicrobial activity against *S.* Gallinarum, whose MIC50 was determined. (**b**) The solution, loaded to Sep-Pak column, was used to perform flow-through (FT), followed by washing twice (WT) with 0.05% trifluoroacetic acid solution, and eluted by 10% to 80% acetonitrile (CAN), and finally the fractions were also used to determine the MIC50. The *y*-axis and different letters (A, B, C, D and E) in the graphs represent the MIC50 and significant differences (*p* < 0.05), respectively.

**Figure 6 toxins-12-00432-f006:**
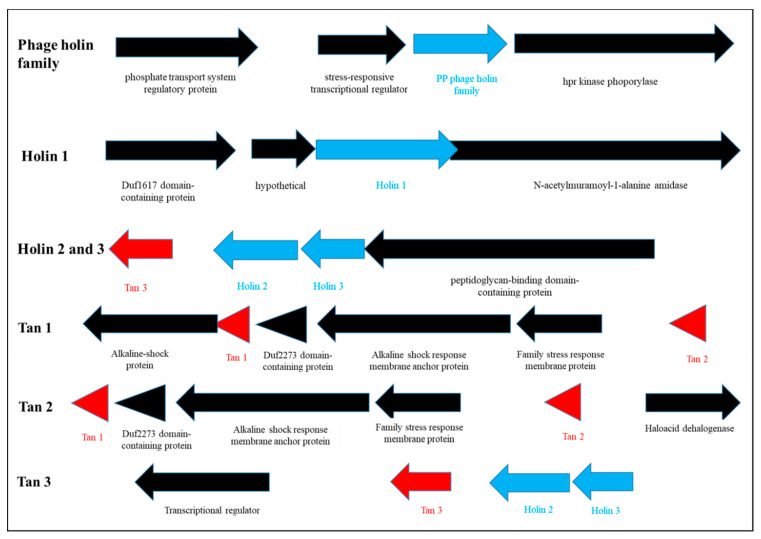
Location of *AMP* genes in *L. taiwanensis* genome. The seven *AMP* genes identified in this study have various flanking genes. Each gene was marked by arrows, whose head is indicated to the orientation of gene expression. In particular, the novel *AMP* genes are displayed by red arrows.

**Figure 7 toxins-12-00432-f007:**
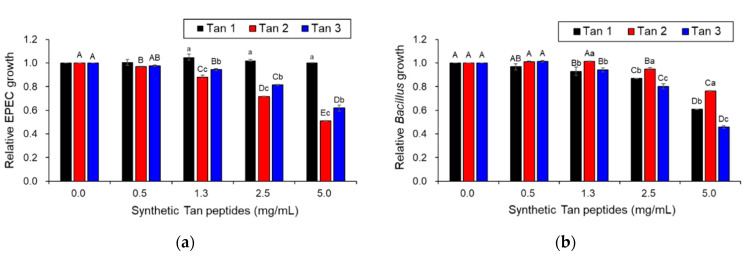
Antibacterial activity of Taiwanencins identified from the *L. taiwanensis* genome. Tan 1, 2, and 3 were synthesized as peptides comprising the amino acid sequences FAWKNRRKIQEVLNHGQS (Tan 1), IVGGVLNLAGGALIMAAGIVKLASKTK (Tan 2), and VGAVSLIAWLAMWV (Tan 3), respectively, which were further examined for their antimicrobial activity against a Gram-negative bacterium (EPEC) (**a**) and a Gram-positive bacterium (*B. cereus*) (**b**). The *y*-axis and different letters (A, B, C, D, E, a, b and c) in the graphs represent the relative growth of pathogenic bacteria and significant differences (*p* < 0.05), respectively. (**c**) Scanning electron micrographs of EPEC treated with synthetic Tan 2 (5 mg/mL). EPEC was treated without (the left image) or with synthetic Tan 2 for 16 h (the right image). The red arrows indicate the pores forming in the EPEC membrane.

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
