# Peer review of "Potential of Bacteriocins from Lactobacillus taiwanensis for Producing Bacterial Ghosts as a Next Generation Vaccine"

_toxins, 2020, doi:10.3390/toxins12070432_

Round 1

Reviewer 1 Report

General comments:

This work investigated the antimicrobial substances produced by Lactobacillus taiwanensis and found that several AMPs secreted from this lactic acid bacterium have potential for producing bacterial ghosts as a next generation vaccine. The cell-free supernatant from L. taiwanensis showed antimicrobial activity against a broad range of pathogenic bacteria, such as a livestock pathogen (S. Gallinarum), a pathogenic E. coli, a human foodborne pathogen (B. cereus) and a fish pathogen (S. iniae). Furthermore, the whole-genome sequencing of L. taiwanensis revealed that this strain has diverse bacteriocins confirmed to function as AMPs, and among them are three novel bacteriocins (Tan 1, Tan 2, and Tan 3).

I believe this research is interesting and provides very useful information for the development of new strategies to produce bacterial ghosts. The study was well-designed and had appropriate methodology.

However, I do not agree with the authors when they refer that “To the best of our knowledge, this report is the first that addresses these issues.” (lines 53-54 of page 2). In a brief search, I found at least 2 different studies that use a cell-penetrating peptide based strategy to produce bacterial ghosts: (1) Palm-Apergi C, Hällbrink M. A new rapid cell-penetrating peptide based strategy to produce bacterial ghosts for plasmid delivery. J Control Release. 2008;132(1):49‐54. doi:10.1016/j.jconrel.2008.08.011; (2) Mangoni ML, Papo N, Barra D, et al. Effects of the antimicrobial peptide temporin L on cell morphology, membrane permeability and viability of Escherichia coli. Biochem J. 2004;380(Pt 3):859‐865. doi:10.1042/BJ20031975. The differences of the present study to the existing ones should be presented in the Introduction and reinforced throughout the manuscript.

The manuscript is well-written and clearly presented. There are only a few minor aspects that I would like to see corrected/completed.

Minor comments:

  1. a) Results (lines 64, 85, 99 and 146; pages 2-6): I prefer shorter headings in the Results sections rather than statements;
  2. b) Results (lines 68-73; page 2): give details about the Material and Methods used to determine the peptide content of the cell-free supernatant (μg peptides/mL);
  3. c) Results (line 191; page 6): the information in Table 1 is not relevant for the discussion, so I suggest that it be transferred to Supplementary Materials;
  4. d) Results (lines 185-186; page 5): change the units from “mg peptides/mL” to “µg peptides/mL” to be consistent with the remaining text;
  5. e) Discussion (lines 215-216; page 7): What reason(s) do the authors find for the higher antimicrobial activity of the AMPs against Gram-negative bacteria than Gram-positive ones (Figure 1)?

Reviewer 2 Report

This manuscript covers an extensive range of data. The authors need to significantly revise the presentation of several data sets as the figures are very difficult to read and interpret.  They are not clear, as described below.

Other comments are as follows:

Lines 29-31   This opening statement uses reference 1 which is 20 years old.  This is too old to support the statement.  A later citation should be given.

Lines 29-30   What strain of E.coli do the authors mean is pathogenic?  Also, what pathogenic strain did they use in all of their studies?  This must be specified everywhere.

Line 46          Appears to be incomplete

Figure 1a & b state the amount of the peptide(s) in ug/mL but the Methods section has no information about how these concentrations were determined.

Line 72          It is incorrect to state that the cell-free supernatant has an “extremely low antifungal activity” when only one yeast has been tested.  A yeast is not a fungus and so the manuscript should be revised throughout to correctly discuss testing against a yeast and not a fungus.

Figure 2        How much/what concentration of supernatant/peptides was used to treat the bacteria as shown?

Figure 3        The letters a, b and c have been used 3 times in 3 different ways in this figure, which is confusing.  An alternative way of labelling the figure should be provided.

Figure 3        I found this to be a very confusing data set and the legend very unclear.  Perhaps the authors should divide the figure into two, one dealing with proteinase K and the second with EDTA, for clarity of the results shown.

Figure 3        There is no control information for the heat treatment alone.  How do the authors know that the results in lanes b and c were not due to the heat treatment alone?  Was lane a heat treated as well?

Figure 4        The legend to this figure is too long and detailed.  It should be modified.  It does not need to repeat the Methods.

Figure 4        I could not understand the data presented in this figure.  It is far too complex and confusing, especially with the letters used to signify growth.  I recommend that this figure be entirely reworked and an alternative, clear presentation provided.

Line 146 et seq       Has the determined gene sequence been deposited in GenBank?  This must be done for the publication to be considered further.

Line 150        The length of the genome stated here does not agree with that in lines 331-332.  Please correct or explain the difference.

Figure 5        I recommend that this be separated into 2 figures as 5a is almost impossible to read as shown

Line 173        Should be flanking, not franking

Line 176 et seq       The authors make a significant assumption here that the genes they have cloned have been correctly expressed in E.coli, their products are secreted and that they have the antimicrobial effects shown in Figure 6a and b.  However, no direct evidence has been provided that this is the case.  For example, SDS page gel analysis of the supernatants could be carried to show that the relative cloned proteins have been expressed.  How do the authors know if any other E.coli excreted proteins, other than those which were cloned, have any AMP effects?

Figure 6a & b          Please revise as recommended for other figures, as described above

Materials and Methods

No details are included about how the concentration of the peptides was determined (see comment above) or how the SDA PAGE was carried out.  These should be corrected.

Line 315 et seq       Details of how the primary preparation of DNA was performed must be included

Line 328        What is Quiver?
